Service humanoid robotics: a novel interactive system based on bionic-companionship framework

http://orcid.org/0000-0002-1011-9676 Yang Jiaji 1 JYang@cardiffmet.ac.uk
http://orcid.org/0000-0003-2644-9888 Chew Esyin 1
http://orcid.org/0000-0003-0677-4421 Liu Pengcheng 2
1 Cardiff School of Technologies, Cardiff Metropolitan University , Cardiff, Cardiff , United Kingdom
2 The Department of Computer Science, University of York , York , United Kingdom
Zhang Qichun
Electronic publication date: 2021 Aug 13
Publication date: 2021
Volume: 7
Electronic Location ID: e674
Received 2021 Apr 19; Accepted 2021 Jul 22
Copyright: © 2021 Yang et al.
Copyright year: 2021
Copyright holder: Yang et al.
License: This is an open access article distributed under the terms of the Creative Commons Attribution License, which permits unrestricted use, distribution, reproduction and adaptation in any medium and for any purpose provided that it is properly attributed. For attribution, the original author(s), title, publication source (PeerJ Computer Science) and either DOI or URL of the article must be cited.
License URL: https://creativecommons.org/licenses/by/4.0/

Keywords: Humanoid robotics, Human-robot interaction, Social robotics

Funding: The authors received no funding for this work.

==============================
At present, industrial robotics focuses more on motion control and vision, whereas humanoid service robotics (HSRs) are increasingly being investigated and researched in the field of speech interaction. The problem and quality of human-robot interaction (HRI) has become a widely debated topic in academia. Especially when HSRs are applied in the hospitality industry, some researchers believe that the current HRI model is not well adapted to the complex social environment. HSRs generally lack the ability to accurately recognize human intentions and understand social scenarios. This study proposes a novel interactive framework suitable for HSRs. The proposed framework is grounded on the novel integration of Trevarthen’s (2001) companionship theory and neural image captioning (NIC) generation algorithm. By integrating image-to-natural interactivity generation and communicating with the environment to better interact with the stakeholder, thereby changing from interaction to a bionic-companionship. Compared to previous research a novel interactive system is developed based on the bionic-companionship framework. The humanoid service robot was integrated with the system to conduct preliminary tests. The results show that the interactive system based on the bionic-companionship framework can help the service humanoid robot to effectively respond to changes in the interactive environment, for example give different responses to the same character in different scenes.

Introduction

Humanoid service robots (HSRs) have seen a sharp rise in adoption recently and are seen as one of the major technologies that will drive the service industries in the next decade (Harris, Kimson & Schwedel, 2018). An increasing number of researchers are committed to investigating HSRs to help humans complete repetitive or high-risk service and interactive tasks such as serving patients with infectious diseases, delivering meals and so on. Delivery robots, concierge robots, and chat robots have been increasingly used by travel and hospitality companies (Ivanov, 2019). Although the contribution of these achievements mainly comes from the rapid development of robotics engineering, Ivanov et al. (2019) indicated that future research focus will gradually shift from robotics engineering to human-robot interaction (HRI), thus opening up interdisciplinary research directions for researchers.

In the early days, Fong, Thorpe & Baur (2003) proposed that in order to make robots perform better, the robot needs to be able to use human skills (perception, cognition, etc.) and benefit from human advice and expertise. This means that robots that rely solely on self-determination have limitations in performing tasks. The authors further propose that the collaborative work between humans and robots will be able to break this constraint, and research on human-robot interaction has begun to emerge. Fong, Thorpe & Baur (2003) believe that to build a collaborative control system and complete human-robot interaction, four key problems must be solved. (1) The robot must be able to detect limitations (what can be done and what humans can do), determine whether to seek help, and identify when it needs to be resolved. (2) The robot must be self-reliant and secure. (3) The system must support dialog. That is, robots and humans need to be able to communicate with each other effectively. However, dialog is restricted at present. Through collaborative control, dialog should be two-way and require a richer vocabulary. (4) The system must be adaptive. Although most of the current humanoid service robots already support dialog and can complete simple interactive tasks, as propounded in the research, such dialog in the present time remains limited and “inhuman.” In the process of interacting with robots, humans always determine the state of the robot (the position of the robot or the action the robot is doing) through vision, and then communicate with the robot through a dialog system. However, HSR cannot perform this yet as they do not seem to fully satisfy the two-way nature of dialog. Therefore, this research responds to the current gap and attempts to differ from the current HRI research. This research attempts to introduce deep learning into the existing dialog system of HSR, thus advancing the field.

With the continuous development of humanoid robots, more and more humanoid robots are used in the service industry, especially the hospitality industry. Human-Robot Interaction (HRI) has become a hot potato by more and more researchers (Yang & Chew, 2020). However, with the deepening of research, some researchers found that when humans interact with humanoid service robots (HSRs), humans hope that HSRs should have the ability and interest to interact with the dynamic thoughts and enthusiasm of the partner's relationship, and can recognize the environment, blended with what others think is meaningful and the emotions to express sympathy (Yang & Chew, 2020). This coincides with Trevarthen’s (2001) companionship theory, so the concept of human robot companion (HRC) was proposed this research. The earlier concept of the robot companion is mentioned by Dautenhahn et al. (2005): HSRs need to have a high degree of awareness and sensitivity to social environment. Through the review of the above literatures, it is proposed to establish an interactive and companion framework for HSRs using deep learning and neural image caption generation, thus advance the current field of HSRs to tackle with bionic-interactive tasks of the service industry and further evolve from conventional HRI to Human and Robot Companion (HRC) (See Table 1).

Table 1 Scenario based comparison for HRI and HRC.

Scenario:	HRI	HRC	
Scenario 1: Hospitality:
The enhancement from HRI to HRC:
(1) Compared with HRI, robots in HRC can recognize the environment (luggage) and changes in customers’ appearance (red shirt), which is in line with the proposed concept of companion should become dynamic in the theory of companionship.
(2) More enthusiastic and bionic interaction capabilities (automatically detect whether they are regular customers, and greet enthusiastically).
(3) The robot in HRC remember the customers’ past orders and provide meal recommendation for well-being.	When you enter a hotel, you see a reception area dominated by robots. When you approach the reception area, the HSR will say “Welcome to Hotel XYZ, please follow the instructions to check-in on my display screen”. After completing the check-in, the robot will tell you the room number and issue you a room card, you go to your room, change a suit and prepare to go downstairs to eat. When you go back to the reception, the robot says ‘welcome, please follow the instructions to place an order on my display’. You choose a few dishes that look good on the screen of the robot, but when the food comes up you don't seem to be satisfied with the taste…	When you enter a hotel, you can a reception area thoughtfully served by robots. The robots also see you and wave to you, ‘Welcome Jack, you have a nice luggage, I can help you to check-in. What else can I do for you?’After completing the check-in, the robot will tell you the room number and issues you a room card. You go to your room and change to a red shirt to go downstairs to eat. When you go back to the reception, the robot says, “Welcome Jack, you wear nice red shirt, what can I do for you?” You choose a few dishes that look good on the robot’s screen, but the robot tells you that ‘According to your past order and diet preferences, these meals may not be suitable for you. Feel free to change it to a less cholesterol dishes with special house promotion and I recommend you to take this quality wine as a treat to have a healthy eating while enjoying your stay with us.’	
Scenario 2: Health care:
(1) The enhancement from HRI to HRC:
Robots have dynamic thinking and real-time neural image captioning ability: able to deal with emergencies and a quick decision making from what it sees the environment in real-time.
(2) Robots has been improved from conventional smart Q&A and interactions to new concept of bionic companionship.	You bought a robot at home to monitor your health. The robot obtains some of your health indicators (such as temperature, blood pressure, etc.) through some external devices. When there is a problem with your indicators, the robot can give you corresponding suggestions or help you contact a doctor. A total of 1 day you suddenly fainted at home for some reasons, but because you did not aim at the detection device connected to the robot, the robot did not find your condition. Fortunately, your neighbor found you fainted at home. . .	You bought a home care robot to monitor your health. The robot obtains some of your health indicators (such as pulses, blood pressure, etc.) through some external devices. When there is a problem with your indicators, the robot can give you corresponding suggestions or help you contact a doctor. A total of 1 day you suddenly fainted at home for some reason. The robot discovered your real-time condition through the deep learning vision system and contacted the your family member or hospital in time, subject to what the robot sees, e.g., fainted human with lots of blood or motion (call hospital for emergency); fainted human with conscious and free speech (call family members).	

This study proposes that the introduction of visual data into the current HRI model of HSRs enables HSRs to have a high level of sensitivity to the social dynamic environment while interacting with humans, thereby enhancing the current HRI model to HRC. With the continuous development of deep learning, some researchers have recently realized the transformation of static pictures or videos from conventional camera input into text descriptions (Li et al., 2020; Hu et al., 2020; Luo, Hsu & Ye, 2019). This deep learning algorithm model is called neural image capturing (NIC). This research attempts to adapt and integrate NIC into HSRs and propose a novel framework (bionic-companionship framework) to enhance the traditional HRI experience. This framework aims to improve the current HRI interaction mode in the field of HSRs to a higher level of HRC (Yang & Chew, 2021). The bionics in this research refers to the humanoid service robot imitating all the tastes of life, trying to adapt to the seven emotions of ancient human nature (joy, anger, sadness, fear, love, disgust, liking) and six biological wills (life, death, eyes, ears, mouth, nose) (Chew et al., 2021). The system proposed in this study combines visual intelligence and Speech Intelligence, and imitates human behavior in social activities, which is in line with the concept of robot bionics proposed by researchers such as Chew et al. (2021). Therefore, this study believes that the proposed system is a bionic system.

Related works

With the continuous development of HRI research, industrial robots have been able to interact with humans accurately and self-adaptively. Some advanced control systems (Zhang, Hu & Gow, 2020) and algorithms (Tang, Zhang & Hu, 2020) have been proposed as Industrial robots provide reliable support for completing interactive tasks in an industrial environment. However, as HSRs began to enter the service industry, some research cases began to discover that there are still problems with the interaction of HSRs in the social environment. Caleb-Solly et al. (2018) believed that users can also help robots when robots help users; meanwhile, users can give feedback to optimize the system. The feedback reflects not only the optimization of the robot system but also the satisfaction of customers. Chung & Cakmak (2018) study indicated that hotels in the hospitality industry want to collect customer feedback in real-time to immediately disseminate positive feedback and respond to unsatisfactory customers while they are still on the scene. Guests want to inform their experience without affecting their privacy. Stakeholders in the hospitality industry hope that intelligent robots can interact more with users. Besides, Rodriguez-Lizundia et al. (2015) concluded that the optimal distance between users and robots is 69.58 cm. Specifically, interaction with a certain greeting mode can attract users to maintain a longer interaction time; robots with the active search are more attractive to participants. The interaction time is longer than that of passively searching robots, suggesting that robots should be designed to keep at a certain distance from humans and consider adding the ability to allow robots to actively identify customers and attract them.

Research suggests that the current interactive system used by HSRs lacks the ability to process and adapt to dynamic social environments. The dynamic social environment here refers to the same human behavior and language often expressing different meanings in different social situations, such as In different situations, the handshake may require two completely different interactive messages to respond. Therefore, this research proposes the concept of HRC to develop a new interactive mode to solve the current problems faced by HRI in the hospitality industry. For a more detailed comparison of HRI and HRC, please refer to the video in the appendix link (https://youtu.be/fZmV4MKeYtQ).

Review of neural image captioning

The challenge of generating natural language descriptions from visual data has been extensively researched in the field of computer vision. However, early research has mainly focused on generating natural language descriptions from video-type visual data (Gerber & Nagel, 1996; Mitchell et al., 2012). These systems convert complex visual data into natural languages using rule-based systems. However, because the rules are artificially designed, these systems are sufficiently robust, bionic, and have been shown to be beneficial in limited applications such as traffic scenarios (Vinyals et al., 2015). In the past decade, various researchers, inspired by the successful use of sequence-to-sequence training with neural networks for machine translation, proposed a method for generating image descriptions based on recurrent neural networks (RNNs) (Cho et al., 2014; Sutskever, Vinyals & Le, 2014). In fact, this method of replacing the encoder in the encoder-decoder framework in machine translation with image features transforms the original complex task of generating image data caption into a simple process of “translating” the image into a sentence (Cho et al., 2014). Furthermore, Donahue et al. (2014) used long short-term memory (LSTM) for end-to-end large-scale visual learning processes. In addition to images, Donahue et al. (2014) also applied LSTM to videos, allowing their models to generate video descriptions. Vinyals et al. (2015) and Kiros, Salahutdinov & Zemel (2014) initially proposed the structure of a currently popular neural image generation algorithm based on the combination of a convolutional neural network (CNN) image recognition model and a natural language processing (NLP) structured model. Moreover, the neural image captioning algorithm based on the attention mechanism has also attracted extensive attention in the field of computer vision. Denil et al. (2012) proposed a real-time target tracking and attention recognition model driven by sight data. Tang, Srivastava & Salakhutdinov (2014) proposed an attention-generation model based on deep learning. From the perspective of visual neuroscience, the model requires object-centric data collection for model generation. Subsequently, Mnih et al. (2014) proposed a new recurrent neural network model, which can adaptively select specific areas or locations to extract information from images or videos and process the selected area at high resolution. As the algorithm has increasingly mature, the application of the algorithm in related fields has also been breaking through recently, such as the caption generation of car images (Chen, He & Fan, 2017), the description generation of facial expressions (Kuznetsova et al., 2014), and educational NAO robots driven by image caption generation for video Q&A games for children’s education (Kim et al., 2015). Recent research on image caption generation also shows that the accuracy and reliability of the technology have increased (Ding et al., 2019). In addition, reinforcement learning to automatically correct image caption generation networks have also been proposed (Fidler, 2017). These deep learning–based studies have undoubtedly laid a foundation for the possible NIC integration with HSRs as proposed in this study. The novel integration led to the possibility for humanoid robots to interact with humans while recognizing the social environment in real time, thereby improving the interactive service quality of the HSRs.

Neural image caption generation algorithm ‘crash into’ robot

An increasing number of studies have been conducted on HRI combined with image caption generation algorithm. Kim et al. (2015) used the structure of a convolutional neural network (CNN) combined with RNN + deep concept hierarchies (DCH) to design and develop an educational intelligent humanoid robot system for play video games with children. In this study, CNN was used to extract and pre-process cartoons with educational features, and RNN and DCH were used to convert the collected video features into Q&A about cartoons. During the game, after watching the same cartoon, the child and the robot ask and answer questions based on the content of the cartoon. The research results show that such a system can interact effectively with children. However, for HRIs, such simple and limited-structured Q&A conditions cannot satisfy all the interaction scenarios required. Cascianelli et al. (2018) used a gull-gated recurrent unit (GRU) encoder-decoder architecture to develop a human-robot interface that provides interactive services for service robots. This research solves a problem called natural language video description (NLVD). The authors also compared the performance when using LSTM and GRU with two different algorithms to solve these problems. They demonstrated that the GRU algorithm runs faster and consumes less memory. This type of model may be more suitable for HSRs. Although the research model is competitive on public datasets, the experimental results on the designed datasets show that the model suffers from significant overfitting. This proves that in the actual model training process, a specific training dataset for HSR interaction should be established, and other methods (such as transfer learning) should be considered to improve the generalization ability of the model for interactive tasks. Luo et al. (2019) created a description template to add various image features collected by the robot, such as face recognition and expression, to the generated description. Compared with the previous models, their interaction is slightly more natural and closer to the human description. However, Luo et al. (2019) use the model to provide limited services to industry managers, hard to generalize, and not for developing an entire HRI framework.

Like the research on robot vision language, research on robot vision action is in its infancy. Yamada et al. (2016) used RNNs to enable robots to learn commands online from humans and respond with corresponding behaviors. This research furthermore provides a reference and direction for humanoid robots to use deep learning to obtain online learning capabilities for human commands. Inspired by the above study, the rationale and hypothesis proposed in the present research are that the description generated by the neural image captions can drive HSRs to perform appropriate behaviors, and HSRs can even obtain online learning capabilities of interacting with surrounding people through studying and analyzing social environments. Tremblay et al. (2018) and Nguyen et al. (2018) believe that non-experts often lack the rationality of task descriptions when issuing instructions to robots. They use deep learning to allow robots to automatically generate human-readable instructions’ descriptions according to the surrounding social environment. In addition, Nguyen et al. (2018) also used visual data to make humanoid robots imitate and learn human actions under corresponding commands so that the robot can learn how to complete the corresponding tasks only through visual data; however, social robots cannot complete precise control of movements when they imitate movements of visual data.

Contribution to the knowledge: the bionic-companionship framework with nic for hsrs

The contribution of the present study is the novel investigation and design of the bionic-companionship framework for HSRs, adapting and integrating neural image caption generation algorithms and bionic humanoid robots, to be validated in a lab-controlled environment and real-life exploration. The new HRC framework is anticipated to enhance HRI to reach a new state, making it possible for HSRs to become bionic companions of humans.

This study proposes adapting and integrating deep learning techniques to one of the world’s most advanced HSRs so that robots can autonomously and in a timely fashion convert pictures or data information captured by robotic visions and sensors into texts or sentences in order to respond and communicate more naturally with humans. The conceptual model of the proposed system consists of various modules, as shown in Figs. 1, 2. The contributions of this research are summarized as follows:

Figure 1 HSR capabilities with the proposed high-level of HRC conceptual model.

Figure 2 Bionic-companionship framework design.

In order to solve the current problems of HSRs in the hospitality industry, a new interactive concept-HRC is proposed.

A novel bionic interaction framework is designed based on the proposed HRC.

A system that can be used on HSRs is developed based on the bionic interaction framework, and the system has been tested and verified. The preliminary results prove that the system can enable HSRs to handle dynamic social environments.

Humanoid service robot used in research

The design and investigation of this HRC framework involves using the Canbot U05E humanoid robot (see Fig. 1 for the high-level design, Figs. 2–5 for further details) (CANBOT, 2020). The robot’s 22-degree-of-freedom motion joints enable it to perform a variety of simulated movements, such as raising the head, turning the head, raising the arm, shaking the crank, shaking hands, leaning back, walking, and turning, and based on the proposed framework, it can acquire natural human behaviors and, as a result, efficiently interact with humans. In addition, Canbot U05E’s advanced vision system and sensors can collect more complete environmental data for the proposed design and make the novel framework more robust. The robot is designed to imitate the human’s seven senses, providing strong support for the concept and implementation of the bionic partner designed in this study.

Figure 3 Neural image captioning model structure for HSR.

Figure 4 Samples Flickr 8k (Rashtchian et al., 2010) training data set.

Figure 5 The infrastructure of the humanoid service robot generating neural image captions as part of the bionic companionship framework.

Bionic-companionship framework

In this study, we review the previous works on this topic and research gaps in the literature and describe a novel humanoid service robot and human interaction framework with neural image subtitles as its core (details are shown in Fig. 2). The framework uses the structure of the NIC algorithm to better realize the interaction of HSRs from HRI to the direction of bionic-companionship. According to the initial descriptions of robot companions, as in the studies by Turkle (2006) and (Kim et al., 2015), the proposed framework should provide HSRs with more natural interactions and a more sensitive understanding of the environment, and hence, the design of the framework is divided into two subsystems (see the dotted red).

Image/video description generation system

These subsystems are the core modules of the entire interactive framework. HSRs collect visual data of the surrounding environment through equipped visual sensors (such as HD or 3D cameras) and sensors (such as tactile and radar). The type of visual data collected depends on the complexity of the interactive task to be completed by HSRs. It is generally considered that more complex interactive tasks require the use of continuous images or real-time videos. The system uses the latest neural image generation algorithm structure and CNN to perform feature extraction on the pictures and video data of the surrounding social environment, and converts the data into feature vector sequences that can be used by RNN. Finally, the RNN completes the process of generating an interactive description from the visual data. HSRs use a speech synthesis system that converts these descriptions into voices to communicate with humans. This process is different from the past mode of using HSRs human sensing sensors and setting fixed interactive feedback; the innovation of this system is that HSRs can automatically and naturally generate interactive feedback. This means that the change in the scene during the interaction will cause a continuous change in the interaction feedback, and this change is not preset by humans. In addition, in further conversation interactions, human voice response and social environment data will be coordinated by HSRs and produce continuous conversation interaction behavior.

Command-robot behavior system

For HSRs, simple conversation interactions are insufficient. HSRs should generate corresponding motions based on visual and human behavior data. For example, when humans wave to a robot, the robot should also actively respond. The hypothesis of this study is to classify or cluster description text generated from visual data and use these classified description texts to control the motions of HSRs in response to complex interactive tasks. For example, when the description generated by neural image captions is “Hello”, then HSRs will automatically determine whether ‘Hello’ matches a category that requires interactive motion and performs corresponding motions such as waving.

Pilot testing, preliminary results, and discussion

In the present study, we designed and integrated a classic NIC model on the HSR and performed a preliminary evaluation.

Introduction to HSR-NIC model

The structure of the HSR-NIC algorithm used in this study was adapted and enhanced from the model structure proposed by Mao et al. (2014) who used a classic encoder-decoder structure. In this study, the encoder uses the Xception pre-trained CNN to convert the input image into a feature vector. The word sequence is then input into the LSTM after a layer of word embedding layer, and finally, and add operation is performed on the word features output by the LSTM and the image features extracted by the trained CNN. These are then input into a decoder composed of a single-layer fully connected layer, which generates the probability distribution of the next word using a softmax layer. The LSTM introduced by the model can solve the long-term dependency problem in the traditional RNN, thereby improving the accuracy of the model. The dense representation of word embedding can reduce the amount of calculations involved in the model; it also enables the model to capture similar relationships between words. In addition, the model used in this study also introduces a dropout layer with a probability of 50% to increase the robustness of the model. The teacher forcing mechanism was used during model training to accelerate the model training process. The optimizer used in the research is Adam, which has the advantages of making the model converge more quickly and automatically adjusting the learning rate with learning. The variables of the model are updated by minimizing the cross-entropy loss between the probability distribution of the predicted result and the probability distribution of the true result and back-propagation. The model structure diagram as follow (Fig. 3).

Model forward propagation process

The training process of the image captioning task can be described as follows: For a picture in the training set, its corresponding description is a sequence that represents the words in the sentence. For model θ, given input image I from the HSR’s vision, the probability of the model generating sequence is expressed as

(1) P(S|I;θ)=Πt=0NP(St|S0,S1,…,St−1,I;θ)

The logarithm of the likelihood function is used to obtain the log-likelihood function:

(2) log⁡P(S|I;θ)=∑t=0N⁡log⁡P(St|S0,S1,…St−1,I;θ)

The training objective of the model is to maximize the sum of the log-likelihoods of all training samples:

(3) θ∗=argmaxθ∑(I,S)⁡log⁡P(S|I;θ)

where (I, S) is the training sample. This method of maximum likelihood estimation is equivalent to empirical risk minimization using the log-loss function. Therefore, in the forward propagation process of this research model, the image feature vector Iv is extracted from the image using the CNN, and a two-dimensional vector of shape (batch size, 2048) is the output.

(4) Iv=CNNθc(I)

The extracted image features need to be encoded by a fully connected layer into the context feature vector C that can be matched with word features. The word feature vector is the output Ot of the LSTM over the time step. The input word of LSTM passes through a word-embedding layer to generate a dense vector representation W(s).

(5) C=Wθ(Iv),Ot=LSTMθ(W(s))

Finally, word feature Ot and context feature C are together input into a decoder composed of a single fully connected layer after the softmax calculation generates the probability distribution of the next word P(Si|I;θ).

(6) P(Si|I;θ)=softmax(Wθ(C+Ot))

The loss function is expressed as

(7) L=∑t=1T⁡y(t)log⁡p(t)+(1−yt)log⁡(1−pt)

Training dataset

For the present study, we use Flickr 8k (Rashtchian et al., 2010) as the training dataset. This is a new benchmark collection for sentence-based image descriptions and searches. It consists of 8,000 images. Each image was paired with five different captions. These captions provide content descriptions of the objects and events in the picture. The images do not contain any well-known people or locations but depict random scenes and situations. Examples of datasets are shown in Fig. 4. The Flickr 8k dataset not only contains images of animals and objects, but also of some social scenes. These data can help robots to better understand natural, day-to-day scenes.

The process of humanoid service robot generating image captions

To explore the feasibility of the bionic-companionship framework, preliminary tests were conducted on a real humanoid service robot (Canbot U05E). The process of generating image captions by a humanoid service robot is divided into four steps, as shown in Fig. 5.

Step 1. The HSR-NIC API is responsible for controlling the robot to call the high-definition camera to collect surrounding environment information (the data collection in this study is focused on HSR capture images). The collected data will be sent to the local host service program through the HTTP protocol and wait for a response from the HSR.

Step 2. The HSR-NIC localhost server program receives the data, and the requests perform preliminary processing and cleaning of the data (image) and send the data (image) to the HSR-NIC model server program to wait for the calculation result (the generated caption description).

Step 3. The HSR-NIC model server program analyzes the image data according to the training parameters saved before, generates the descriptive caption, and returns it to the local server.

Step 4. The HSR-NIC local server program sends the caption description to the robot application through the HTTP protocol, and the robot application controls the robot to respond according to the caption description, such as speech synthesis and motion control.

Preliminary test results and limitation

In this study, we conducted a preliminary test on a humanoid service robot integrated with the NIC algorithm. The results of the preliminary test were found to be promising.

With the discuss of the last chapter, the research will integrate the NIC into the HSRs to make the HSRs take advantage of the change of the surrounding environment interact with the human better. Therefore, the system proposed by this research will combine qualitative analysis and quantitative analysis to initially validate the performance of the system.

This study introduces the cross-entropy loss curve of the last 50 epochs of the model as the evaluation metric for quantitative analysis. As shown in the Fig. 6, the model finally converges to the minimum loss value of 2.65 in the training set and 2.71 in the validation set, which proves that the model has no over-fitting and under-fitting, and has generalization ability. Since the loss value is calculated from the sum of the difference between the probability value of each predicted word in the predicted description and the true value, the loss value will be affected by the sentence length of the predicted description. In related work, researchers (Li et al., 2020; Hu et al., 2020) used some more reliable evaluation methods to evaluate the performance of the model, including the BLUE4 (Papineni et al., 2002) and CIDEr (Vedantam, Lawrence Zitnick & Parikh, 2015). These evaluation metrics are usually used in the field of machine translation instead of manual evaluation. Since the tasks handled by the NIC model can be regarded as translated from images/scenes into English, the evaluation metrics can also be applied to the evaluation of NIC. This study will use qualitative analysis to replace quantitative analysis of metrics such as BLUE4 and CIDEr, so as to further evaluate the preliminary performance of HSR after the integrated NIC model.

Figure 6 Loss curve of NIC model on training set and validation set.

As shown in Figs. 7 and 8, the researcher conducted two sets of tests in three different scenarios with HSR. In the first set of tests, the researcher wore a hat and changed scenarios. In the second set of tests, the researcher did not wear a hat, and the scene switching method was the same as in the first set. It can be seen from the experimental results that the humanoid robot can complete the perception of scene switching through this algorithm and generate a rough description of the scene. In the first set of tests, most of the content described was accurate. The robot equipped with the NIC algorithm can effectively identify ‘man’, ‘black shirt’, and ‘sitting on a bench’. However, in the second group of tests, there were many errors in the recognition results. This could be attributed to the researcher’s long hair. Interestingly, researchers with long hair are easily identified as women or children. This indicates that the accuracy of the NIC algorithm still has room for improvement.

Figure 7 A series of preliminary testing results captured from Canbot U05E and bionic-companionship preliminary framework.

Figure 8 Real social environment test examples.

In addition, in order to test the performance of the system in a dynamic environment. The researcher conducted the test in a real environment (as Fig. 8). The researcher selected six real environments as the test data and let the robot generate interactive information. Among the six real interactive environments, there are three scenes that can be more accurately recognized by the robot and produce corresponding descriptions. The description information can correspond to the test environment, and the corresponding part of the description has been highlighted with the same color in the Fig. 8. Some of the objects, facilities, and human movements in these scenes can be accurately predicted, such as sidewalk, traffic, bench, building, building, etc. However, in the other three environments, the robot did not give an accurate description. The researchers believe that this may be due to the fact that the training set does not contain objects in these three environments, causing the model to fail to learn how to express the ‘unfamiliar environment’.

In general, as per the results of the to two experimental sets, it was proven that the robot equipped with the NIC algorithm can capture the changes in the surrounding environment and generate different feedbacks according to the changes. The results also demonstrate the feasibility of the proposed bionic-companionship framework. Although there is still a gap between the prediction results of the algorithm and the real communication scene, the researcher believes that special data collection for some specific interaction scenarios and model training for these specific data can be effective in addressing this gap. Future research directions will mainly focus on improving the accuracy of algorithms and achieving more human-like interactions. (The detailed process is shown in the HSR-NIC demo video.) In addition, the researcher believes that the scene understanding of static images is the basis for dealing with dynamic environments. Some researches have mentioned that the introduction of related algorithms of object detection into NIC can identify and generate descriptions of scenes in dynamic environments. This is also the current research limitation of this research and the research challenges that will be faced in the future.

Conclusions

This study presents a review of neural image generation algorithms and application cases in the field of robotics, and proposes a novel humanoid service robot and human interaction framework based on the bionic-companionship theory. The subsystems of the bionic-companionship framework are designed and introduced in detail. Preliminary tests also initially proved that the framework could increase the sensitivity of HSRs to changes in the surrounding environment. The proposed framework will contribute to further development from HRI to HRC. Future work will focus on implementing each of the subsystems in the framework and applying the framework to HSRs to verify its performance.

Supplemental Information

Supplemental Information 1 The key code of the novel interactive system.

Click here for additional data file.

Supplemental Information 2 Demo video for system run.

Click here for additional data file.

Additional Information and Declarations

Competing Interests

Author Contributions

Data Availability

Esyin Chew & Pengcheng Liu are Academic Editors for PeerJ.

Jiaji Yang conceived and designed the experiments, performed the experiments, analyzed the data, performed the computation work, prepared figures and/or tables, authored or reviewed drafts of the paper, and approved the final draft.

Esyin Chew conceived and designed the experiments, prepared figures and/or tables, authored or reviewed drafts of the paper, and approved the final draft.

Pengcheng Liu conceived and designed the experiments, prepared figures and/or tables, authored or reviewed drafts of the paper, and approved the final draft.

The following information was supplied regarding data availability:

The research code is available in the Supplemental File. The training dataset for the system is Flickr 8k and is available at Kaggle: https://www.kaggle.com/adityajn105/flickr8k.

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
