# Peer review of "Service humanoid robotics: a novel interactive system based on bionic-companionship framework"

_PeerJ Computer Science, doi:10.7717/peerj-cs.674_

## Round 0.1 · original submission · Major Revisions

Based on the reviewers' comments, a major revision is needed to improve the current manuscript where more experimental results should be added and the reference list should be enriched. Generally speaking, the paper is well-written with publishable contents however careful proofreading is necessary as an essential part of revision.

Reviewer 1 ·

Basic reporting

The paper proposed a novel interactive system based on the bionic-companionship framework. On the basis of ensuring the dynamic environment sensitivity of the interactive system in the humanoid service robot, the neural image capture technology in deep learning is adopted to improve the experience experiment of human-computer interaction. The results show that the proposed method improves the system performance to a certain extent. The writing structure is reasonable, clearly organized, the diagrams and formulas are standardized. Some minor problems of this manuscript should be revised.
1. More newest references should be cited.
2. The English writing should be polished

Experimental design

The experimental design is resonalbe, and the results and analysis are convicing.

Validity of the findings

The proposed method is innovative.Conclusions are well stated.

Additional comments

The paper proposeD a novel interactive system based on the bionic-companionship framework. On the basis of ensuring the dynamic environment sensitivity of the interactive system in the humanoid service robot, the neural image capture technology in deep learning is adopted to improve the experience experiment of human-computer interaction. The results show that the proposed method improves the system performance to a certain extent. The thesis is somewhat innovative. The writing structure is reasonable, clearly organized, the diagrams and formulas are standardized, and the experimental results and analysis are sufficient. Some minor problems of this manuscript should be revised.
1. More newest references should be cited.
2. The English writing should be polished

Reviewer 2 ·

Basic reporting

The paper is well presented with a clear structure. The literature references are sufficient in the field. The raw data and source codes have been shared.

Experimental design

The research question is well defined. The study aims to improve the current HRI interaction of the HSRs by integrating a deep learning-based NIC model into the system. The general system framework has presented clearly in the method section. However, as the crucial component (model) in the system, more details related to the NIC are desirable. E.g. the training protocol used in the experiments, the way to combine the visual cues of CNN and textual cues in the LSMT for image captioning.

Validity of the findings

The authors provided some qualitative analysis and preliminary tests in the paper to validate the initial performance of the proposed HRI framework. However, those testing cases should have been well defined as a part of experiments. E.g. it would be better that the cases/scenarios could be explicitly designed within a controlled environment, e.g. covering different scenes, human/object, indoor/outdoor environment, etc. Moreover, the validation should include some quantitative analysis in the study.

Additional comments

This paper presented a general bionic-companionship framework for HSRs to enhance HRI, where a classic NIC model was introduced into the system with other models (e.g. motion, language/voice, emotion). The paper is easy to read with a clear structure and written. Specific suggestions related to the method and analysis are as follows:
• As the NIC model is the critical component in the proposed framework, more details related to the NIC should be described, e.g. Optimizer, training protocol, fusion strategy etc.
• Although the authors have presented some preliminary tests in this study, the design of testing cases needs to be described and justified. Meanwhile, both quantitative and qualitative analyses are desirable to validate the framework. The NIC model could be validated separately before deploying to the entire system.
• The abbreviation NRC presents in line 65 should be clarified, Human-Robot Collaboration (HRC)?

·

Basic reporting

This paper touches upon a very attractive and promising area of human-robot-interaction which could be a potent, cross-disciplinary research domain in future robotics. In its current form, the paper need to be significantly improved. I would suggest the authors to take my comments into consideration, and address my concerns.

Basically, the presentation of this paper is partially ambiguous on its originality and significance, which I will point out later in details.

The literature covers well the main focus of this paper. However, it is insufficient to be inclusive for readers/researchers from wider communities. Besides that, the authors need to double-check the format of references.

Generally speaking, this paper is well organised and written. However, the figures presentation is absolutely the shortcoming of this research paper. The authors should make great efforts for such improvement.

The differences between this paper and a recent conference paper (with a very similar topic) by the authors upon any further investigation, or perspectives, or new methods, contributions...should be more obviously highlighted in the Abstract and Introduction.

The validation of the proposed results is reasonable as to my understanding. Some details are yet missing to articulate the main results.

Experimental design

As to me, the experimental design should comprise more cases to verify the interactive system. I can only pick up one effective testing result which is casually presented. In other words, I feel the experiment is incomplete. I would like to see any further, interesting result from the robot actions when interacting within the stated "social dynamic environment".

The method has been simply, clearly formulated which looks reasonable and correct to me.

Validity of the findings

The main contribution should be more highlighted. In the present presentation, I read several times at some sections that I still find some difficulty to grab quickly the novelty and contribution of this paper. The authors should improve it to be easy to follow.

Additional comments

General comments:
1. Basic Perspective
This paper touches upon a very attractive and promising area of human-robot-interaction which could be a potent, cross-disciplinary research domain in future robotics. In its current form, the paper need to be significantly improved. I would suggest the authors to take my comments into consideration, and address my concerns.

Basically, the presentation of this paper is partially ambiguous on its originality and significance, which I will point out later in details.

The literature covers well the main focus of this paper. However, it is insufficient to be inclusive for readers/researchers from wider communities. Besides that, the authors need to double-check the format of references.

Generally speaking, this paper is well organised and written. However, the figures presentation is absolutely the shortcoming of this research paper. The authors should make great efforts for such improvement.

The differences between this paper and a recent conference paper (with a very similar topic) by the authors upon any further investigation, or perspectives, or new methods, contributions...should be more obviously highlighted in the Abstract and Introduction.

The validation of the proposed results is reasonable as to my understanding. Some details are yet missing to articulate the main results.

2. Experiment and Method
As to me, the experimental design should comprise more cases to verify the interactive system. I can only pick up one effective testing result which is casually presented. In other words, I feel the experiment is incomplete. I would like to see any further, interesting result from the robot actions when interacting within the stated "social dynamic environment".

The method has been simply, clearly formulated which looks reasonable and correct to me.

3. Contribution
The main contribution should be more highlighted. In the present presentation, I read several times at some sections that I still find some difficulty to grab quickly the novelty and contribution of this paper. The authors should improve it to be easy to follow. Listing crucial points one by one is an usual way to summarise the significance and improve the readability.

Detailed comments:
This paper proposes an attractive idea on designing an interactive system in service humanoid robots by introducing the widely applied neural image capturing in deep learning. The robot, in this case, could respond effectively to dynamic social environment.

I have the following concerns and comments in details:

Major concerns:

1. A major concern is about the statement of "bionic-companionship". As to me, deep learning or called deep neural learning is not a fully bionic pathway to Artificial Intelligence. Here the term "bionic" may mislead authors. To my understanding, bionics, more or less, should draw some inspirations from biological systems for some embodiments in engineering systems and technology. I did not see very clear evidence for this point. How would your service robot behave, or interact with humans resembling a biological entity?

2. Figure presentation needs to be SIGNIFICANTLY changed and improved. Even when I zoom in, I cannot see clearly the embedded information (Fig. 5). Could you also replace Fig. 3 with a more structural model illustration on NIC ? Fig. 1 could be separated into two sub-figures.

3. Related work is insufficient. Considering convincing readers from wider communities, it is better to see some additional points:
(1) Research milestones in HSRs
(2) Differences (including SoTA systems, and challenges) between the research focusing on industrial robotics and HSRs which have been stressed in the Abstract
(3) Emergence of challenges and methods when bridging HRI to HSR

4. Experiments - It is good to see the interesting, preliminary testing results in Fig. 6. However, how does this result reflect your statement of dealing with social dynamic environment?

5. Clarifying the significance - I found some repetitions with the authors' recent conference paper. What is the main extension of this journal paper? Either deeper/more systematic investigation or new proposed method?

Small recommendations:

1. Abstract:
(1) "The problem and quality of human-robot interaction (HRI) has become a widely debated topic in academia." - Could you extend it a bit more to say what exactly the major problem is?
(2) Could you emphasise more the main contribution DIFFERENTLY to previous works?
(3) "changes in the interactive environment" - ambiguous, what kinds of changes?

2. Introduction:
(1) "high-risk service and interactive tasks" - for some examples?
(2) Maybe I missed some info, what is the "HRC" herein short for?
(3) How do the authors define the "social dynamic environment", empirically? or with any standard? This is important as to be addressed by the proposed system.
(4) I would like to see the organization of the rest of this paper at the end of Introduction.

3. I would like the authors to elaborate on the methods with more details.

---

## Round 0.2 · accepted · Accept

After the major revision, the quality of the manuscript has been improved significantly while the contribution in the revised version has been highlighted. Based on the reviewers' comments, I would like to recommend accepting this manuscript as it is.

Reviewer 2 ·

Basic reporting

The authors have significantly improved the manuscript according to the reviewer's comments. I'm happy with the authors' responses to my comments, including clarification of the NIC model as well as its evaluation with more testing cases and scenarios.

Experimental design

The research question is well defined. The experimental design is justified with concise implementation. More descriptions are added in the revision for clarification.

Validity of the findings

The method evaluation section has been improved significantly, more testings of method robustness are added, and both quantitative and qualitative analyses are presented. The conclusion can be drawn from the reported results.

Additional comments

There are many improvements based on the reviewers' comments in this version. I'm happy with the authors' responses and corresponding revisions to my comments.

·

Basic reporting

The authors have carefully revised the manuscript regarding my suggestions. My major concerns on Figure representations have been also well addressed. The quality in its current form is acceptable.

Experimental design

The experiments are clear to show the advantages. I found the supplementary video is very helpful to understand the main work.

Validity of the findings

This work is technically sound.

Additional comments

I would appreciate the consideration of my recommendations for improving this paper in its previous form. The authors have made great efforts on revising this paper into details. My major concerns on the figure representations have been addressed. The major and minor points have been considered, enhanced and clarified. I found the supplementary video very helpful for the understanding of main work. I do NOT have further points for another round of revision. Therefore, I would fully support the publication of this research paper.